# Glassy thermal conductivity in Cs₃Bi₂I₆Cl₃ single crystal

Paribesh Acharyya[1], Tanmoy Ghosh[1], Koushik Pal [2,7], Kewal Singh Rana[3], Moinak Dutta[1], Diptikanta Swain [4], Martin Etter [5], Ajay Soni [3], Umesh V. Waghmare [2,6] & Kanishka Biswas [1,6] ✉

As the periodic atomic arrangement of a crystal is made to a disorder or glassy-amorphous system by destroying the long-range order, lattice thermal conductivity, $\kappa_L$, decreases, and its fundamental characteristics changes. The realization of ultralow and unusual glass-like $\kappa_L$ in a crystalline material is challenging but crucial to many applications like thermoelectrics and thermal barrier coatings. Herein, we demonstrate an ultralow (~0.20 W/m·K at room temperature) and glass-like temperature dependence (2–400 K) of $\kappa_L$ in a single crystal of layered halide perovskite, Cs₃Bi₂I₆Cl₃. Acoustic phonons with low cut-off frequency (20 cm⁻¹) are responsible for the low sound velocity in Cs₃Bi₂I₆Cl₃ and make the structure elastically soft. While a strong anharmonicity originates from the low energy and localized rattling-like vibration of Cs atoms, synchrotron X-ray pair-distribution function evidence a local structural distortion in the Bi-halide octahedra and Cl vacancy. The hierarchical chemical bonding and soft vibrations from selective sublattice leading to low $\kappa_L$ is intriguing from lattice dynamical perspective as well as have potential applications.

Quantized normal modes of lattice vibration, called phonons, are the primary carrier of thermal energy in the crystalline insulators. The strength of phonon scattering, which blocks the propagation of thermal energy, varies from material to material depending on the intrinsic (chemical bonding and crystal structure) and extrinsic (solid solution point defects, nano/microstructures, etc.) parameters. Crystalline materials at both extreme end of ultrahigh or ultralow lattice thermal conductivity ($\kappa_L$) are technologically relevant to numerous applications[1]: while low thermal conductive materials are beneficial for thermoelectrics[2,3] and thermal barrier coatings[4], high thermal conductive materials are needed for heat transmission and dissipation[5].

According to kinetic theory $\kappa_L = 1/3 C_v v_g l_{ph}$, where $C_v$ is the volumetric heat capacity, $v_g$ is the phonon group velocity, and $l_{ph}$ is the phonon mean free path. Materials have low $\kappa_L$ either because of comprising strong phonon scattering centers (e.g., strong lattice anharmonicity[6–9], rattling dynamics[10–12], excess disorder[13,14], hierarchical nanostructuring[15]) and/or because of low $v_g$ due to soft chemical bonding and heavy atomic masses[16]. The wave nature of phonons, however, sets an intrinsic limit to the lower value of $\kappa_L$ ($\kappa_{min}$): $l_{ph}$ cannot be lower than the half of the wavelength ($\lambda/2$: $\lambda$ is the phonon wavelength) or the interatomic spacing ($a$) of the lattice. In crystalline insulators, $\kappa_L$ decreases with increasing temperature above the Debye temperature ($\theta_D$) and $\kappa_{min}$ is asymptotically reached in the highly anharmonic regime at high temperatures (T ≫ $\theta_D$). As the strength of anharmonicity increases, and/or the unit cell becomes large and complex, and/or disorder increases the temperature regime at which

[1]New Chemistry Unit, Jawaharlal Nehru Centre for Advanced Scientific Research (JNCASR), Jakkur P.O., Bangalore 560064, India. [2]Theoretical Science Unit, Jawaharlal Nehru Centre for Advanced Scientific Research (JNCASR), Jakkur P.O., Bangalore 560064, India. [3]School of Basic Sciences, Indian Institute of Technology Mandi, Mandi, Himachal Pradesh 175075, India. [4]Institute of Chemical Technology-IndianOil Odisha Campus, Bhubaneswar 751013, India. [5]Deutsches Elektronen-Synchrotron (DESY), 22607 Hamburg, Germany. [6]School of Advanced Materials and International Centre for Materials Science, Jawaharlal Nehru Centre for Advanced Scientific Research (JNCASR), Jakkur P.O., Bangalore 560064, India. [7]Present address: Department of Materials Science and Engineering, Northwestern University, Evanston, IL 60208, USA. ✉e-mail: kanishka@jncasr.ac.in

$\kappa_L \sim \kappa_{min}$ decreases[16,17]. Disorder is particularly notorious in limiting $l_{ph}$: in fact, in amorphous materials and glasses, where the long-range periodicity of a crystal lattice is destroyed, $l_{ph} \sim \lambda/2$ (or, $l_{ph} \sim a$) is the highest frequency and the only remaining well-defined available phonon mode throughout the entire temperature range. This limit is also often called the glass limit. Thermal conductivity of amorphous materials and glasses then has significantly lower value compared to their crystalline counterparts, and it slowly increases with increasing temperature with an asymptotic limit of $\kappa_{min}$, in drastic contrary to the crystalline insulators[18,19].

Crystalline materials in which $\kappa_L$ reaches this glass limit and exhibit glass-like temperature variation of $\kappa_L$ ($\kappa_L(T)$) is extremely important for a variety of applications such as in thermoelectrics. Previously glass-like $\kappa_L$ has been observed in crystalline materials when they comprise large and complex unit cell[20–22], incorporate large amount of disorder[23] and nanostructures[24,25], or have specially designed layered modules[9]. However, the presence of disorder is often unwarranted as it deteriorates charge carrier transport. The challenge is then how to realize glassy $\kappa_L$ in a crystalline material, in the absence of impurity and disorder. Such materials would not only be fascinating from the perspective of chemical bonding and lattice dynamics but would have immense technological importance.

In this work, we demonstrate such glassy thermal conductivity in a large single crystal and high symmetry (space group: $P\text{−}3m1$) structure of all-inorganic metal halide perovskite $Cs_3Bi_2I_6Cl_3$. Recently, metal halide perovskites have received unprecedented attention because of their fascinating physical and chemical properties[26–28]. The outstanding electronic and optoelectronic properties and their application are, however, found to be greatly affected by their lattice vibrations (phonons)[29,30]. Investigations of thermal conductivity of these halide perovskites are essential for application as it influences their stability, operating lifetime of a device and long-term device performance[31,32]. Although the influence of lattice vibrations on electronic and optical transitions is to some extent explored, thermal transport properties of halide perovskites in the presence of a temperature gradient are still in its infancy[33,34]. Only a handful of studies have been carried out investigating the thermal conductivity of these halide perovskites[35–37], particularly, for the all-inorganic halide perovskites[38–43].

Herein, we have demonstrated ultralow lattice thermal conductivity ($\kappa_L$) in a Bridgman grown large single crystal of layered $Cs_3Bi_2I_6Cl_3$ perovskite. The $\kappa_L$ value is found to be 0.20 and 0.22 W/m·K at room temperature when measured perpendicular and parallel to the Bridgman growth directions, respectively. The temperature (2–400 K) dependence of $\kappa_L$, both for the parallel and perpendicular to the Bridgman growth directions, exhibits the behavior similar to that of amorphous materials and glasses. At a very low temperature (<10 K), the temperature dependence of $\kappa_L$ in $Cs_3Bi_2I_6Cl_3$ strongly deviates from the typical $T^3$ dependence of crystalline materials. Second, in the intermediate temperature range, the temperature dependence of $\kappa_L$ lacks the peak found in typical crystalline materials. Rather, it exhibits a flat plateau in the temperature range (40–150 K), and then a gradual increase with temperature approaching the glass limit. First-principles density functional theory (DFT) calculations of the harmonic interatomic force constants clearly reveal the large difference in bond stiffness of Bi−I/Cl and Cs−I/Cl, and demonstrate very soft elastic moduli, corroborating our experimentally measured low sound velocity in $Cs_3Bi_2I_6Cl_3$. In accordance, its phonon dispersion exhibits soft acoustic phonons with a rather low cut-off frequency (20 cm⁻¹) and reveals an abundance of weakly dispersive low-energy optical phonon modes below 60 cm⁻¹, which have been further verified using low-temperature heat capacity (2–50 K) and temperature-dependent (4–300 K) Raman measurements. Synchrotron X-ray pair distribution function (PDF) analysis revealed the presence of a local structural distortion in the

Bi-halide octahedra of $Cs_3Bi_2I_6Cl_3$ which resulted in local symmetry breaking. Thereby, the combined presence of soft crystalline lattice; low energy optical phonons and high lattice anharmonicity due to atomic rattling; local distortion and vacancy results in low phonon lifetime and consequently in an ultralow and glass-like temperature dependence of $\kappa_L$ in $Cs_3Bi_2I_6Cl_3$.

## Results and discussion

### Crystal structure

$Cs_3Bi_2I_6Cl_3$ (space group: $P\text{−}3m1$) is a new member in the $\langle 111 \rangle$ oriented layered halide perovskites family and it is isostructural to $Cs_3Bi_2Br_9$ and $\alpha\text{-}Cs_3Sb_2I_9$[44,45]. Each octahedron of $Cs_3Bi_2I_6Cl_3$ comprises three capping iodine atoms that terminate the bilayers and three bridging chlorine atoms that connect the octahedra to adjacent opposite layers, thereby forming a layered structure with 2D connectivity (Fig. 1a). The bridging Cl atoms in $Cs_3Bi_2I_6Cl_3$ make it a 2D structure when compared to 0D $Cs_3Bi_2I_9$ which are devoid of such Cl atoms[44].

Large single crystal of $Cs_3Bi_2I_6Cl_3$ (Fig. 1b; for details see "Methods", Supplementary Information, SI) has been synthesized by Bridgman crystal growth. The crystal structure of $Cs_3Bi_2I_6Cl_3$ is solved and refined using single crystal X-ray diffraction at room temperature (Supplementary Tables 1, 2). Figure 1c exhibits the typical Laue diffraction spots collected from the $Cs_3Bi_2I_6Cl_3$ single crystal at room temperature. The laboratory XRD (Cu Kα) patterns of $Cs_3Bi_2I_6Cl_3$ crystal, measured at room temperature for both parallel and perpendicular cut to the Bridgman growth directions, are shown in Fig. 1d. The crystallographic c-axis is perpendicular to the Bridgman growth direction, i.e., the stacking direction of the layers is perpendicular to the Bridgman growth direction (Supplementary Fig. 1). The room temperature synchrotron ($\lambda = 0.7762$ Å) powder XRD pattern of the finely ground sample could be also indexed with the trigonal space group ($P\text{−}3m1$) as shown in Supplementary Fig. 2a. The obtained lattice parameters $a = b = 8.2438(3)$ Å and $c = 10.0337(6)$ Å, from the refinement, also agree well with the lattice parameters obtained from our single crystal XRD data (see Supplementary Table 1)[44,45].

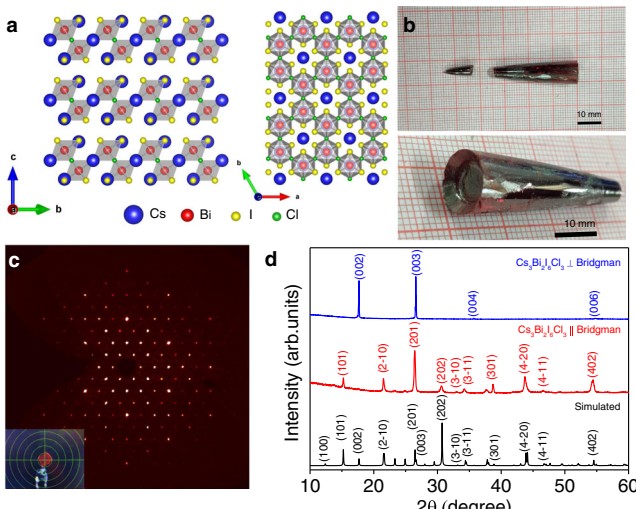

**Fig. 1 | Crystal structure of $Cs_3Bi_2I_6Cl_3$. a** Crystal structure of the $\langle 111 \rangle$ oriented layered halide perovskite $Cs_3Bi_2I_6Cl_3$. **b** Optical photographs of the Bridgman grown $Cs_3Bi_2I_6Cl_3$ single crystal. **c** Laue diffraction spots (along c direction) from the $Cs_3Bi_2I_6Cl_3$ single crystal. Inset shows photograph of a broken piece of single crystal specimen mounted on the sample holder. **d** Room temperature XRD pattern of $Cs_3Bi_2I_6Cl_3$ single crystal cut along parallel (∥) and perpendicular (⊥) to the Bridgman growth direction.

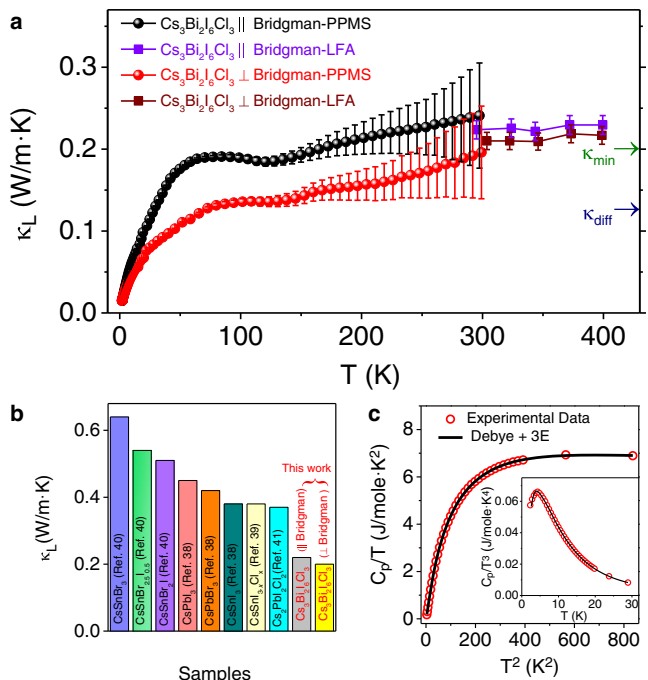

**Fig. 2 | Glass-like thermal conductivity in $Cs_3Bi_2I_6Cl_3$. a** Temperature-dependent lattice thermal conductivity ($\kappa_L$) of $Cs_3Bi_2I_6Cl_3$ along parallel (||) and perpendicular (⊥) to the Bridgman growth directions measured using PPMS (2–300 K) and LFA (300–400 K). $\kappa_{min}$ is theoretical minimum value of $\kappa_L$ and $\kappa_{diff}$ is diffusive thermal conductivity. **b** Comparison of room temperature $\kappa_L$ of $Cs_3Bi_2I_6Cl_3$ single crystal with other low thermal conductive all-inorganic halide perovskites[38–41]. **c** Low-temperature $C_p/T$ vs. $T^2$ plot of $Cs_3Bi_2I_6Cl_3$ and fit with Debye-Einstein model. Inset shows $C_p/T^3$ vs. $T$ plot exhibiting Boson-like peak in low-temperature $C_p$.

## Glass-like thermal conductivity

Thermal conductivity of $Cs_3Bi_2I_6Cl_3$ in the temperature range of 2–400 K along parallel (||) and perpendicular (⊥) to the Bridgman growth direction is shown in Fig. 2a. Thermal conductivity is measured using a DynaCool physical property measurement system (PPMS) and a laser flash apparatus (LFA 457) in the 2–300 K and 300–400 K temperature range, respectively (Fig. 2a). Because of the insulating nature of the sample with band gap ~2.0 eV (Supplementary Fig. 4), the total thermal conductivity has a negligible contribution from the charge carriers and, thus, can be considered nearly equal to the lattice thermal conductivity ($\kappa_L$). $\kappa_L$ exhibits a slightly lower value for the entire measured temperature range along the ⊥ to Bridgman growth direction (i.e., || to the crystallographic *c*-direction) when compared to that of the || to Bridgman growth direction (i.e., ⊥ to the *c*-direction) (see schematic in Supplementary Fig. 1). This observed thermal anisotropy is because of the stacking of 2D $[Bi_2I_6Cl_3]^{3-}$ layers along the crystallographic *c*-direction.

We observed several intriguing features in the experimentally measured thermal conductivity of $Cs_3Bi_2I_6Cl_3$ in the 2 to 400 K range. The $\kappa_L$ at 300 K along ⊥ and || to the Bridgman directions has ultralow values ~0.20 and ~0.22 W/m·K, respectively, which are close to the theoretical minimum value of $\kappa_L$, $\kappa_{min}$ = 0.19 W/m·K estimated using Cahill's model (Supplementary Table 3)[23]. The estimated value of $\kappa_{min}$ (i.e., $\kappa_{diff}$) according to the diffuson model proposed by Agne et al.[18] is 0.12 W/m·K for $Cs_3Bi_2I_6Cl_3$ (Supplementary Table 3). The measured room temperature value of $\kappa_L$ of $Cs_3Bi_2I_6Cl_3$ is lower compared to that of the other all-inorganic metal halide perovskites (Fig. 2b). Moreover, layered compounds generally exhibit highly anisotropic thermal conductivity between the in-plane and out-of-plane directions (Supplementary Fig. 5a). However, $Cs_3Bi_2I_6Cl_3$ exhibits weak anisotropy with ratio of ~1.10 (based on LFA data) between $\kappa_L$ values along || and ⊥ to

the Bridgman growth directions at 300 K due to the low anisotropy in the measured sound velocities between the in-plane and out-of-plane direction, which have average values of ~1156 m/s and ~1076 m/s along the || and ⊥ to Bridgman growth directions, respectively (Supplementary Table 3). The interlayer separation (~3.57 Å) is shorter than the expected van der Waals distance (~3.96 Å, van der Waals radius of iodine is ~1.98 Å), but longer than the covalent bonding (~2.78 Å, covalent radius of iodine is ~1.39 Å). Although the structure resembles the 2D layered materials without any overlapping charge clouds, weak interlayer interaction exists and results in a weak thermal anisotropy in $Cs_3Bi_2I_6Cl_3$[44]. A similar weak anisotropic thermal conductivity has also been reported recently for an organic-inorganic hybrid perovskite $BA_2PbI_4$ crystal, in which preferential alignment of the organic chains between two inorganic layers along the out-of-plane direction lowers the anisotropy[46]. The third and most significant observation is the unusual glass-like temperature dependence of $\kappa_L$ of $Cs_3Bi_2I_6Cl_3$ despite its crystalline structure and measurement of thermal conductivity using a single crystal specimen. An increasing $\kappa_L$ with increase in temperature has been observed recently in layered $WSe_2$ crystal[13] and $BaTiS_3$[47]. Crystalline materials generally exhibit an ~$T^3$ dependent $\kappa_L$ at low temperature, followed by a peak after which $\kappa_L$ decreases following a $T^{-1}$ dependence due to Umklapp scattering and finally reaches a temperature independent $\kappa_{min}$ value when the phonon mean free path becomes of the order of interatomic distance[23]. However, unlike typical crystalline materials, the temperature dependence of $\kappa_L$ of $Cs_3Bi_2I_6Cl_3$ largely deviates from a $T^3$ (Supplementary Fig. 5b) dependence at low temperature. The temperature dependence of $\kappa_L$ of $Cs_3Bi_2I_6Cl_3$ also lacks crystalline-like peak in the intermediate temperature range. Rather, we observed a plateau region (40–150 K) in the temperature dependence of $\kappa_L$ of $Cs_3Bi_2I_6Cl_3$, which is also typically observed in amorphous materials and glasses[17,48]. After the plateau, a weakly increasing and then nearly constant $\kappa_L$ was observed at high temperature that approaches the glass limit ($\kappa_{min}$).

## Chemical bonding and lattice dynamics

We performed first-principles DFT calculations for the analysis of chemical bonding (Fig. 3a), crystal structure and lattice dynamics of $Cs_3Bi_2I_6Cl_3$ to understand its ultralow and glass-like thermal conductivity. We have performed crystal orbital Hamilton population (COHP) analysis (inset of Fig. 3a and Supplementary Fig. 6) using wavefunctions obtained from DFT. The sharp anti-bonding peaks (near −0.2 eV) just below the Fermi level ($E_f$) are associated with Bi−Cl and Bi−I interactions. Below −0.75 eV, however, Bi−I interactions have a bonding character whereas Bi−Cl interactions maintain anti-bonding character down to −2 eV below $E_F$. Such filled anti-bonding states in the valance bands just below the $E_f$ soften the lattice, leading to low sound velocities. On the other hand, Cs−I and Cs−Cl interactions show nearly vanishing COHP characters, indicating no covalency and primarily ionic interaction between them (inset of Fig. 3a and Supplementary Fig. 6). Analysis of harmonic interatomic force constants (IFCs) clearly reveals the large difference in bond stiffness of Bi−Cl/I and Cs−Cl/I bonds (Fig. 3a) with the Bi−I bond being the strongest (force constant, |Φ| = 5.0 eV/Å²) followed by the Bi−Cl bond stiffness (|Φ| = 1.5 eV/Å²). The disparity in interactions of Cl and I with the metal atoms (Bi and Cs) gives rise to a bonding hierarchy that helps in suppressing $\kappa_L$ in the compound[16]. Chemical interactions between different species can also be rationalized by charge-density analysis as shown in Fig. 3b on a supercell of the crystal structure of $Cs_3Bi_2I_6Cl_3$. It is seen that the charge clouds of Bi and I strongly overlap, confirming their strong covalent bonding. In comparison, the overlap between the Bi and Cl charge cloud is negligibly small. Because of this vanishingly small charge cloud overlap between Bi and Cl having bond distance 2.92 Å and low bond stiffness, the Bi−Cl interaction can be considered as nearly ionic and the $Cs_3Bi_2I_6Cl_3$ structure can also be alternatively described as covalently bonded $BiI_3$ units distributed in an ionic matrix

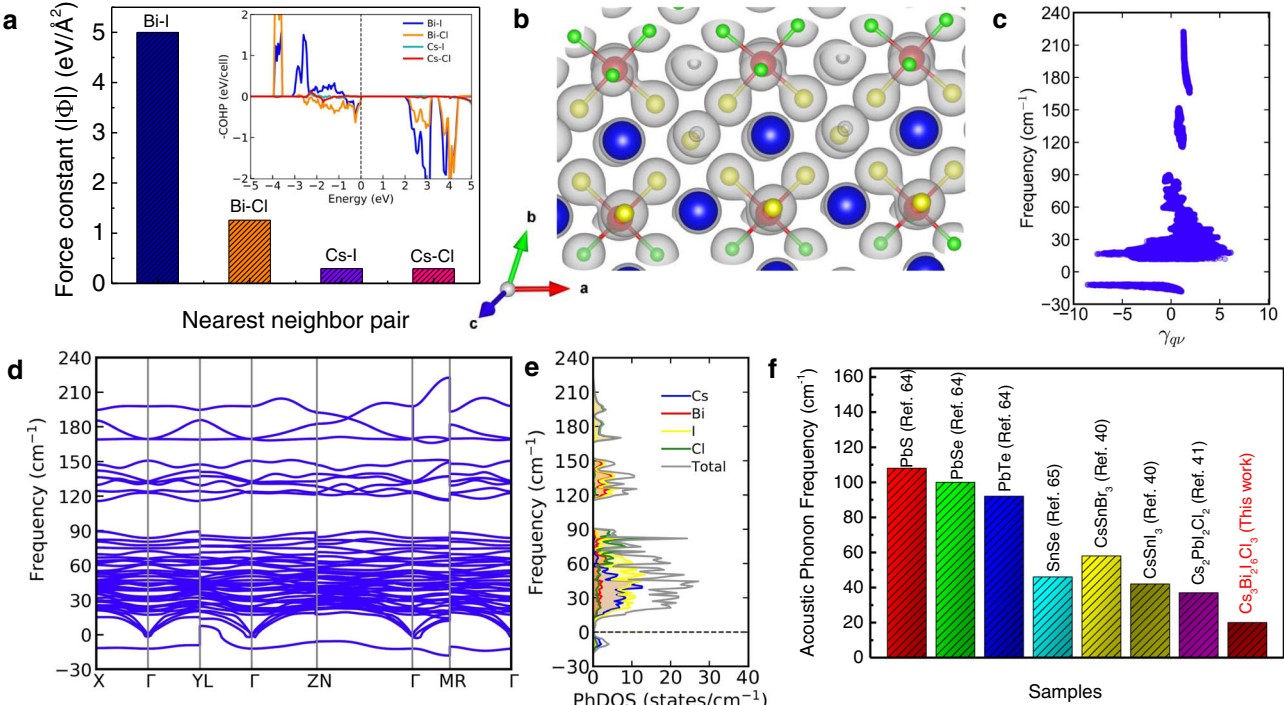

**Fig. 3 | Chemical bonding and lattice dynamics of $Cs_3Bi_2I_6Cl_3$. a** Harmonic interatomic force constants for the nearest neighbor atom pairs in $Cs_3Bi_2I_6Cl_3$. Inset shows the crystal orbital Hamilton population (COHP) analysis of $Cs_3Bi_2I_6Cl_3$. Positive and negative values in the $y$-axis indicate bonding and anti-bonding states, respectively. **b** Total charge-density plot in the supercell of $Cs_3Bi_2I_6Cl_3$, where iso-surfaces of charge-density are shown in light gray color. Isosurface of charge densities has been visualized at an iso-value of 0.04 $e/Bohr^3$. Cs, Bi, I, and Cl atoms are denoted by the blue, red, yellow, and green spheres, respectively. **c** Calculated mode Grüneisen parameters of $Cs_3Bi_2I_6Cl_3$. **d** Phonon dispersion and **e** atom-resolved phonon density of states of $Cs_3Bi_2I_6Cl_3$. **f** Comparison of acoustic phonon frequency of $Cs_3Bi_2I_6Cl_3$ with well-known low thermal conductive heavy metal chalcogenides and 3D all-inorganic halide perovskites[40,41,64,65].

of $Cs^+$ and $Cl^-$. Thus, the presence of bonding hierarchy, different masses of atoms and anti-bonding states induced lattice softening are expected to give rise to strong lattice anharmonicity in $Cs_3Bi_2I_6Cl_3$.

Harmonic phonon dispersion of $Cs_3Bi_2I_6Cl_3$ exhibits multiple imaginary phonon branches (Supplementary Fig. 7a) which is common in the family of halide perovskites[49]. The imaginary phonon mode ($-19\,cm^{-1}$) at $\Gamma$ exhibits a double-well potential energy surface (Supplementary Fig. 7b). To stabilize the structure, we nudged the atoms along the eigenvectors of this imaginary phonon mode at $\Gamma$ and relaxed the crystal structure. Calculated phonon dispersion for this relaxed structure of $Cs_3Bi_2I_6Cl_3$ exhibits a weakly unstable ($-12\,cm^{-1}$) phonon branch with negligible dispersion (Fig. 3d). Phonon dispersion of $Cs_3Bi_2I_6Cl_3$ also exhibits notably soft acoustic phonon modes with cut-off frequencies below $20\,cm^{-1}$ (Fig. 3f) in all directions of the Brillouin zone, which are responsible for the soft elastic moduli and low speeds of sound in this compound (Supplementary Table 3). The calculated bulk and shear moduli of $Cs_3Bi_2I_6Cl_3$ are 14.4 and 6.7 GPa, respectively, which agree well with the values estimated from the experimentally measured sound velocity (Supplementary Table 4). The bulk and shear moduli of $Cs_3Bi_2I_6Cl_3$ are lower compared to most other materials exhibiting ultralow $\kappa_L$ (Supplementary Fig. 8).

The phonon dispersion (Fig. 3d) exhibits multiple low-energy optical phonon branches with high density below $90\,cm^{-1}$. Interestingly, the unstable phonon branch with frequency $-12\,cm^{-1}$ is rather weakly dispersive and bears similarities to a branch of phonons with rattling atoms (Supplementary Fig. 9)[40,50]. The potential energy landscapes of the atoms in $Cs_3Bi_2I_6Cl_3$ shows that the Cs atoms have the shallowest potential energy surface in all the direction compared to other atoms (Bi, I, and Cl) in the crystal structure signifying their rattling-like dynamics (Supplementary Figs. 9, 10). The phonon dispersion (Fig. 3d) and phonon density of states (Fig. 3e) reveal that the unstable phonon mode primarily involve the vibrations of Cs and Cl

atoms. The analysis of participation ratio (PR)[51] of the phonon modes (Supplementary Fig. 11) reveals that the unstable phonon mode near $-12\,cm^{-1}$ have low PR value (-0.2–0.3), indicating its localized nature, which is also the characteristics of a rattling phonon mode[40]. Further, low PR values indicate the signature of several diffuson modes (Supplementary Fig. 11b) in $Cs_3Bi_2I_6Cl_3$. Generally, $\kappa_L$ due to diffuson-mediated heat transport weakly increases with temperature[52,53]. The atom-resolved phonon density of states (PhDOS) (Fig. 3e) also reveals that the contribution of Cs vibrations is mostly localized in a narrow energy window centered around $40\,cm^{-1}$. Because of similar atomic masses of Cs and I, we also observe contributions of iodine atoms to phonon bands centered around $40\,cm^{-1}$. However, due to the strong bonding of I to the lattice, these phonon modes are dispersed over a relatively broader frequency range. Due to the high density of phonons at low energies (~$40\,cm^{-1}$), they can give rise to numerous phonon-scattering processes that can strongly suppress $\kappa_L$[16]. Further analysis shows that the torsional mode distorts the soft $[BiCl_{6/2}I_3]^{3-}$ octahedra due to the presence of a smaller size of Cl at the bridging position, making the crystal structure more flexible and elastically soft (Supplementary Fig. 7).

$Cs_3Bi_2I_6Cl_3$ possesses low elastic moduli (Supplementary Fig. 8 and Supplementary Table 4) resulting in soft phonon frequencies that generally possess strong anharmonicity. To quantify the anharmonicity of the phonon modes in $Cs_3Bi_2I_6Cl_3$, we estimated the mode Grüneisen parameters ($\gamma_{qv}$) (Fig. 3c), which are notably large (≫1) for the acoustic and low-energy optical phonons. The estimated average Grüneisen parameter from experimental sound velocities is -2.3 (Supplementary Table 3). Since phonon scattering rates (inversely proportional to phonon lifetimes) vary inversely with square of $\gamma_{qv}$[54], large values of $\gamma_{qv}$ limits $\kappa_L$ to an ultralow in $Cs_3Bi_2I_6Cl_3$.

We found that the excitation of only acoustic phonons, as described within Debye theory, does not satisfactorily account for the

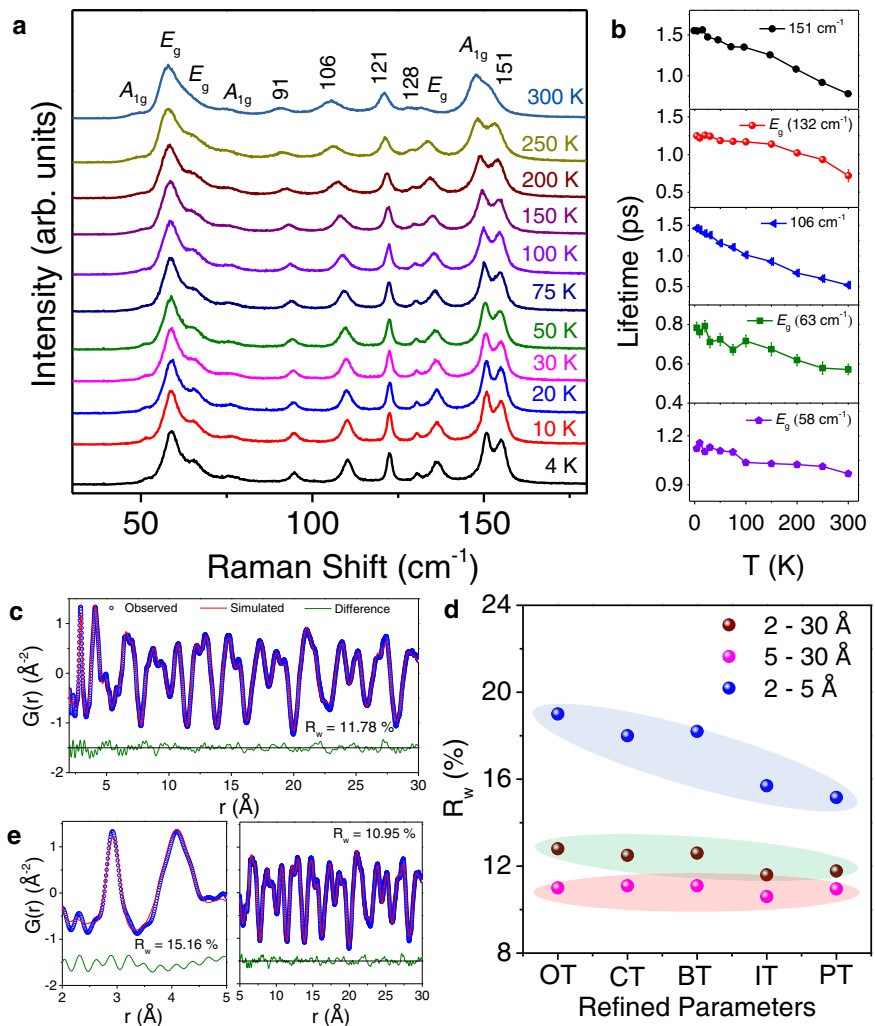

**Fig. 4 | Vibrational characteristics and local structure of $Cs_3Bi_2I_6Cl_3$.**
**a** Temperature-dependent Raman spectra and **b** temperature variation of phonon lifetime for various modes of $Cs_3Bi_2I_6Cl_3$. **c** Room temperature synchrotron X-ray PDF data fitted using $P\bar{3}m1$ space group (ambient crystal structure) with all atom positions and thermal parameters refinement. **d** Goodness of fit ($R_w$) vs. refinement parameters, where OT: only thermal parameters; BT: bismuth position and thermal parameters, CT: caesium position and thermal parameters, IT: iodine position and thermal parameters, and PT: all atomic positions and thermal parameters, are refined. **e** Fitting of local (2–5 Å) and average structural peaks (5–30 Å) for the synchrotron X-ray PDF data of $Cs_3Bi_2I_6Cl_3$ at 300 K using distorted model.

low-temperature heat capacity, $C_p$, of $Cs_3Bi_2I_6Cl_3$ (Supplementary Fig. 12). Rather, a combined Debye-Einstein model (see equation 8, SI) with the inclusion of at least three Einstein modes of characteristics temperatures $\theta_{E_1} = 19.2\,K$ (13.3 cm$^{-1}$), $\theta_{E_2} = 40.9\,K$ (28 cm$^{-1}$), and $\theta_{E_3} = 76.9\,K$ (53.4 cm$^{-1}$) (Fig. 2c and Supplementary Table 5) satisfactorily fit the temperature dependence of $C_p$ in the 2–30 K temperature range. The presence of these Einstein modes could be corroborated with the observation of weakly dispersive low-energy optical phonon modes in our first-principles-based calculation of phonon dispersion in Fig. 3d. These low-energy optical phonon modes could be attributed to the localized vibration of Cs and I atoms (Supplementary Fig. 13). These low energy and localized optical phonon modes result in excess phonon density of states, which could be seen from the presence of a Boson-like peak in $C_p/T^3$ vs. T plot (inset of Fig. 2c) and typically found in amorphous materials and glasses[16,55,56].

## Vibrational characteristics and local structure

The occurrence of low-lying optical phonon modes is further verified by temperature-dependent Raman spectroscopy. Figure 4a shows the temperature-dependent (4–300 K) Raman spectra in the range 30–180 cm$^{-1}$. From group theoretical analysis, 4 $A_{1g}$ and 5 $E_g$ Raman active modes should be present in $Cs_3Bi_2I_6Cl_3$ (Supplementary Fig. 13).

Most of these peaks in the Raman spectra originate from the terminal and bridging symmetric and asymmetric stretching motion of Bi-X (X = Cl and I) bonding[44,57]. We have observed four low-frequency Raman active modes in the range below 80 cm$^{-1}$ with a weak mode positioned at ~48 cm$^{-1}$ ($A_{1g}$), a highest intensive bending mode at ~58 cm$^{-1}$ ($E_g$), ~63 cm$^{-1}$ ($E_g$), and ~75 cm$^{-1}$ ($A_{1g}$) using 785 nm laser. Moreover, symmetrical and asymmetrical stretching modes are observed at ~147 cm$^{-1}$ ($A_{1g}$) and ~132 cm$^{-1}$ ($E_g$) due to the vibration of the octahedra at 300 K (see visualization of the Eigen modes at the Γ point in Supplementary Fig. 13). We have also observed a Raman active mode at ~38 cm$^{-1}$ ($E_g$) using a 633 nm laser and low-frequency filter (Supplementary Figs. 13, 14). The further lower frequency $A_{1g}$ (~20 cm$^{-1}$) and $E_g$ (~22 cm$^{-1}$) modes could not be observed because they are very close to the Rayleigh tail. All the Raman active modes exhibit softening as temperature increases from 4 to 300 K (Fig. 4a and Supplementary Fig. 15b). The integrated intensity (Supplementary Fig. 15a) and FWHM (Supplementary Fig. 15b) of the Raman modes increase with increasing temperature indicating that the phonon population increases and the scattering process intensifies, respectively. We have estimated the phonon lifetime ($\tau_i$) for the Raman active modes by $\tau_i = \frac{1}{2\pi \text{FWHM}_i}$, where FWHM$_i$ is the full-width-half-maxima of the observed peaks[41]. The calculated $\tau_i$ is of the order of pico-seconds (ps) and decreases further

with increasing temperature (Fig. 4b), which reflects strong phonon scattering processes in $Cs_3Bi_2I_6Cl_3$. Furthermore, there are five unassigned Raman modes at 91, 106, 121, 128, and 151 $cm^{-1}$ (at 300 K), which arise probably due to the presence of a local structural distortion in the average structure of $Cs_3Bi_2I_6Cl_3$ single crystal, which would further amplify the phonon scattering strength.

The presence of the symmetry forbidden Raman modes is indicative of structural distortion in $Cs_3Bi_2I_6Cl_3$ and we have carried out temperature-dependent synchrotron X-ray pair distribution function (PDF) analysis to verify this assumption. X-ray PDF is a total scattering technique and gives simultaneous information for both the local and average crystal structure of a material[58]. Figure 4c shows the X-ray PDF data of $Cs_3Bi_2I_6Cl_3$ at 300 K which is fitted using the trigonal crystal structure (space group $P\text{−}3m1$). The peaks below 5 Å provide information about the local structure whereas the peaks above 5 Å describe the average structure of $Cs_3Bi_2I_6Cl_3$ (Fig. 4e). When the X-ray PDF data is refined by tuning only the thermal parameters (OT), we observed that the average structure ($r > 5.0$ Å) can be well described within this model as indicative from its goodness of fit $R_w = 11\%$ (Supplementary Fig. 16). However, the corresponding fit to the local structure ($r < 5.0$ Å) with this structural model is quite poor as evident from high $R_w = 19\%$ (Supplementary Fig. 16). Systematic refinement of atomic positional parameters improves the local structural fitting; however, the best fit is obtained when the positional parameters of all the atoms are refined simultaneously along with their thermal parameter (denoted as PT). The local fit improves significantly to 15.16% for PT as compared to 19% for OT (Fig. 4e). $R_w$ value for the average structure ($r = 5\text{–}30$ Å) remains almost identical in all the cases (Fig. 4d) indicating that the changes are predominantly in the local structure which averages out in the global structure. The Bi−Cl bond distance is 2.92 Å whereas Bi−I bond distance is 2.90 Å in the global structure which is unusual because size of Cl atom is smaller than I atom. The local structural refinement leads to Bi−Cl and Bi−I bond distance 2.92 and 2.94 Å, respectively, leading to a local structural distortion in $[BiCl_{6/2}I_3]^{3-}$ octahedra. The second peak ~4 Å in Fig. 4c corresponds to the Cs and I nearest neighbor correlation. The Cs and I distance is also found to change from 4.08 to 4.02 Å with no notable off-centering of Cs from its parent position. Such local distortion is known to lower the thermal conductivity in few compounds[40,59]. Although the magnitude of local distortion is small but can aid in lowering the thermal conductivity of the material in conjunction with the other phonon scattering phenomena present in this compound. The presence of such a local distortion in the Bi-halide octahedra lowers the local structural symmetry and might be the reason why we observe unassigned extra Raman modes. Moreover, the refinement of the occupancies (Occ) at room temperature indicated Cl vacancy in the material (Occ (Cl): 0.957; Supplementary Table 6) and such vacancy is known to provide unusual behaviors of temperature dependence of thermal conductivity[60]. The atomic displacement parameters (ADPs) are found to be high, mainly for Cs and to an extent for Cl as well (Supplementary Fig. 17), and is in accordance with the theoretical potential energy vs. displacement plot (Supplementary Fig. 10). The high ADP for Cs resembles its rattling character as also observed from the theoretical calculations. Therefore, the combined presence of soft elastic crystal structure, high lattice anharmonicity and the presence of low-frequency optical phonon modes, intrinsic Cl vacancy and local structural distortion in $Cs_3Bi_2I_6Cl_3$ can be attributed to the experimentally observed ultralow and glass-like temperature dependence of $\kappa_L$.

it's worth mentioning here that recent experiments and theoretical studies[42,43,48] have revealed that halide perovskites are extremely anharmonic in nature, manifesting damped vibrations of the octahedral motions, soft phonons, and extremely short phonon lifetimes. The strong lattice anharmonicity of $Cs_3Bi_2I_6Cl_3$ is also evident from our experimental results. Therefore, an accurate description of $\kappa_L$ of $Cs_3Bi_2I_6Cl_3$ requires consideration of temperature-dependent phonon frequency renormalization and inclusion of higher-order anharmonic interactions for the estimation of phonon scattering rates[61,62]. However, the hexagonal symmetry and relatively large unit cell (14 atom) of $Cs_3Bi_2I_6Cl_3$ makes such calculation computationally challenging. Moreover, glass-like $\kappa_L$ is conventionally observed in highly-disordered materials in which disorder limits the phonon mean free path, while few materials recently emerged in which modular crystal structure and strong lattice anharmonicity help mimicking a glass-like temperature dependence[63]. Halide perovskites offer a unique opportunity in which a combined presence of both these effects of structural imperfections (static and dynamic) and lattice anharmonicity could be easily realized[42,43]. This leads to intriguing thermal transport behavior as we have observed in this example of $Cs_3Bi_2I_6Cl_3$, however, it's proper description requires a unified theory of thermal transport which combines both the effects of disorder and lattice anharmonicity within a single theoretical framework[48].

In conclusion, we have demonstrated an ultralow $\kappa_L$ and its glass-like temperature dependence in the 2–400 K range for a Bridgman grown single crystal of 2D all-inorganic halide perovskites $Cs_3Bi_2I_6Cl_3$. The soft elastic layered structure results in a low cut-off frequency (20 $cm^{-1}$) of the acoustic phonons and hence low speeds of sound. Moreover, the chemical bonding hierarchy results in an abundance of weakly dispersive low-energy optical phonon modes, which includes a torsional motion of $[BiI_3Cl_3]^{3-}$ octahedra and localized anharmonic rattling-like vibrations of Cs atoms bonded weakly to the lattice. Soft acoustic phonons, the abundance of low energy optical phonons and high lattice anharmonicity result in strong phonon scattering and short phonon lifetimes (0.5–1 ps at 300 K). A local structural distortion is also evident in the Bi-halide octahedra of $Cs_3Bi_2I_6Cl_3$ by synchrotron X-ray PDF. The combined effect of soft acoustic modes, Cl vacancy, localized anharmonic vibrations, and local structural distortion result in a glass-like temperature dependence of $\kappa_L$ in $Cs_3Bi_2I_6Cl_3$. This intriguing phonon transport with an ultralow $\kappa_L$ and its unusual glass-like temperature dependence in a single crystal of all-inorganic halide perovskite demonstrate a rich interplay between chemical bonding hierarchy and lattice dynamics which could be useful in various optoelectronic and thermoelectric applications.

## Methods

### Synthesis

Single crystal of $Cs_3Bi_2I_6Cl_3$ was grown using a Bridgman furnace by using stoichiometric amount of CsCl and $BiI_3$. The vacuum-sealed ampule kept at 750 °C for 48 h and then moved through a temperature gradient from 600 to 480 °C at a speed of 1 mm/h. Finally, the sample was slowly cooled to room temperature in 120 h.

### Single crystal X-ray diffraction (SCXRD)

The single crystal data were collected at room temperature (298 K) using a Bruker D8 VENTURE diffractometer equipped with a PHOTON detector and graphite-monochromatic Mo-Kα radiation ($\lambda = 0.71073$ Å, 50 kV, 1 mA).

### Thermal conductivity

Thermal conductivity measurement was carried out in the temperature range of 2–400 K. The low-temperature thermal conductivity (2–300 K) was measured using a physical properties measurement system (DynaCool PPMS, Quantum Design). Above room temperature (300–400 K), thermal diffusivity, D, was measured by laser flash diffusivity technique using a Netzsch LFA-457 instrument (Supplementary Fig. 3). The error in thermal conductivity measurement was determined using the standard deviation in thermal conductivity as per the standard protocol used in Quantum Design PPMS. The measurement error for LFA thermal conductivity is 5%.

**Synchrotron X-ray pair distribution function (X-PDF)**

Samples were ground with an agate mortar pestle and then filled in a capillary of 0.6 mm diameter for performing synchrotron X-ray PDF measurements. Both ends of capillaries were sealed using an adhesive. A Perkin Elmer XRD1621 area detector was used to record the diffraction data. The wavelength of the beam was fixed at 0.20742 Å. The data was taken at the P02.1 beamline of PETRA III, DESY, Germany.

**Computational details**

We performed first-principles density functional theory (DFT) calculations using the Vienna Ab-initio Simulation Package (VASP) with potentials derived using the projector augmented wave (PAW). The details of the methods are mentioned in Supplementary Information (SI).

**Reporting summary**

Further information on research design is available in the Nature Research Reporting Summary linked to this article.

## Data availability

The data that support the findings of this study are available from the corresponding author upon reasonable request. The CIF data generated in this study have been deposited in the Cambridge Crystallographic Data Centre under accession code "2150864 [https://doi.org/10.25505/fiz.icsd.cc2b64qf]".

## Code availability

The custom codes used in this work are available under reasonable request.

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

## Acknowledgements

K.B. acknowledges support from a Swarnajayanti fellowship grant, Science and Engineering Research Board (SERB) (SB/SJF/2019-20/06), and Department of Science & Technology (DST) (DST/SJF/CSA-02/2018-19). P.A. thanks the University Grants Commission (UGC) for fellowship. We thank DST for financial support, and JNCASR and SINP for facilitating the synchrotron experiments at the Indian Beamline, PF, KEK. We thank G. Manna for maintaining the Indian Beamline, PF during the experiment. Synchrotron X-ray PDF measurements were carried out at beamline P02.1, PETRA III of DESY, a member of the Helmholtz Association (HGF). Financial support by the DST provided within the framework of the India@DESY collaboration is gratefully acknowledged. A.S. acknowledges SERB (CRG/2018/002197). U.V.W. acknowledges funding from a J.C. Bose National Fellowship of the SERB-DST.

## Author contributions

K.B. conceived the idea and designed the study. P.A., T.G., and K.B. carried out the synthesis, structural and other characterizations, thermal conductivity experiments, and analysis of the data. K.P. and U.V.W. carried out the theoretical calculations. K.S.R and A.S. carried out temperature-dependent Raman study. D.S solved the single crystal structure. M.E. collected and M.D. analyzed the PDF data. All authors contributed to writing and editing the manuscript.

## Competing interests

The authors declare no competing interests.
