## [Peer Review File · Nature Communications]

Glassy Thermal Conductivity in Cs₃Bi₂I₆Cl₃ Single CrystalREVIEWER COMMENTS

Reviewer #1 (Remarks to the Author):

In this paper, Acharyya et al. report an ultralow (0.20 W/m.K) thermal conductivity and glass-like temperature dependence in a layered perovskite material Cs₃Bi₂I₆Cl₃. The authors grew single-crystal samples, measured their thermal conductivities over a broad temperature range (2–400 K) using two methods, and performed theoretical calculations on chemical bonding and lattice dynamics. Moreover, the origin of soft phonon modes was tracked down to the rattling vibration of Cs⁺, by combining Raman and X-ray PDF refinement techniques. Overall, the materials performance presented herein is outstanding, the characterization results are comprehensive, and the discussions are thorough. I think this work will attract broad attention in research areas of thermoelectrics and perovskite materials, and recommend its publication in Nat. Comm. after considering the following suggestions and comments:

1. Sections and subheadings can be added to better organize the text contents.
2. The special role of Cl⁻ in the soft phonon modes and ultralow thermal conduction can be investigated more. As mentioned in the text, 'charge clouds of Cl do not have any overlap with that of Bi' and 'the Bi-Cl bond distance is 2.92 Å whereas Bi-I bond distance is 2.90 Å in the global structure which is unusual'. Along with force constant calculations in Fig. 3A and potential curves in Fig. S7A, the Bi-Cl interaction can be better interpreted as ionic, and Cs₃Bi₂I₆Cl₃ can be understood as BiI₃ molecules distributed in an ionic matrix of Cs⁺ and Cl⁻, better than [Bi₂I₆Cl₃]³⁻ layers (p. 6). The contrast between covalent and ionic bonds as well as between heavy and light atoms may well be a reason for the soft and largely isotropic phonon modes.
3. Several sentences should be double-checked for grammar:
 - a. p. 4: temperature dependence of κ_L in Cs₃Bi₂I₆Cl₃ is strongly deviates from...
 - b. p. 10: ... which could be seen from the presence of a Boson-like peak in C_p/T^3 vs. T plot (inset of Fig. 2C) and typical in amorphous materials and glasses.
 - c. p. 12: The peaks below ($r < 5$ Å) provide information about...

Reviewer #2 (Remarks to the Author):

Crystalline solids with ultralow glass-like thermal conductivity have received great attention in recent years as they are important for example for thermoelectrics. This paper reports ultralow glassy thermal conductivity of a layered halide perovskite single crystal, Cs₃Bi₂I₆Cl₃. The authors have combined many techniques including Bridgman crystal growth, single crystal X-ray diffraction, Synchrotron X-ray Pair Distribution Function, DFT calculations, Raman spectra, and thermal conductivity characterizations, which is much appreciated. For a study reporting glassy thermal conductivity this paper is generally good, but the rationalization of glassy thermal conductivity is not well supported by the authors' claims and discussions. Also, the writing is a bit fragmented, and needs to be improved for a better connection between different sections. Below are some detailed comments for the authors to address before considering for publication.

- 1) The authors report the glassy ultralow thermal conductivity of Cs₃Bi₂I₆Cl₃ as well as many data proving large anharmonicity and local structural distortion, which is good. However, the reviewer did not see a clear connection between glassy ultralow thermal conductivity and large anharmonicity (or local structural distortion). One conclusion made by the authors is that localized anharmonic vibrations and local structural distortion result in glass-like temperature dependence of κ_L in Cs₃Bi₂I₆Cl₃. But what is the evidence to support this claim? The detailed origin of the glass-like temperature dependence probably requires further studies.
- 2) Defects and disorders are common in halide perovskites. The structural analyses on single crystal X-ray diffraction, synchrotron powder X-ray diffraction, and Pair Distribution Function did not include the refinements of site occupancies. Detailed analysis of site occupancies is required given point defects are common in halide perovskites.

3) The phonon lifetime extracted from Raman spectra shows a decreasing trend with rising temperature, which reflects strong phonon scattering processes increasing with temperature. One may expect a decreasing κ with increasing temperature as the phonon lifetime decreases. This result and argument from the authors seem to be conflicting with glassy thermal conductivity data, which show a weak increasing trend with temperature. Can the authors comment on this?

4) According to the point group of $D_{3d}(-3m)$ of $P-3m1$, Raman active modes include $4A_{1g} + 5E_g$. From the spectra in Fig. 4A, only $3A_{1g}$ and $3E_g$ modes are indexed. How about the other $1A_{1g}$ and $2E_g$ modes? Can the authors explain this?

5) From the PDF analysis, it seems that the local distortion is quite small as the change in bond distance is rather small (at the second decimal). The reviewer is a bit suspicious of how such a small distortion in the bond distance would result in a glass-like temperature dependence of κ . Can the authors elaborate in detail on how a local structural distortion would result in enhanced phonon scattering strength and a glass-like temperature dependence of κ ? A poor fit of the synchrotron X-ray PDF data usually indicates structure disorder. The refinement model based on the reported cif file did not consider defects such as interstitials and vacancies, which might be the origin of the glass-like κ . Especially, static disorder is often considered as an important reason for the glass-like κ . The authors could refer to a recent paper (Nat Commun 12, 6709 (2021)) for details on studying subtle structure disorders in simple crystalline structure using advanced crystallographic methods.

6) Can the authors comment on the origin of the unstable phonon modes in phonon calculations? Are they caused by the structure disorder or strong anharmonicity? Can these unstable phonons be removed by applying hydrostatic pressure or a phonon renormalization scheme?

7) The reviewer has a question on the Bridgman grown crystal. As shown in Fig. 1D, in principle, one may expect the $(hk0)$ reflections for the // Bridgman direction if it is a perfect single domain. It is usually difficult to ensure a single domain for the whole large crystal. Can the authors comment on this? Could it be a misalignment during cutting or the crystal being multi-domain?

Reviewer #3 (Remarks to the Author):

This work studies the origin of the glassy-like thermal conductivity in all-inorganic metal halide perovskite $\text{Cs}_3\text{Bi}_2\text{I}_6\text{Cl}_3$ single crystals with a combined experimental and theoretical approach. The single crystal has high quality with size being sufficiently large to give promising κ . The κ is surprisingly low 0.2 W/mK at 300K , and the plateau behavior in κ is indeed interesting and uncommon compared to many other inorganic halide perovskites. In summary, the reviewer thinks the work is interesting, but there are a few technical questions that need to be addressed:

1. The experimental observation in κ is interesting and sound. However, the simulation part based on quasiharmonic approximation may not be sufficient to fully account for the unusual κ . Even if anharmonic calculations (e.g., AIMD) or phonon scattering rate calculations will not be included eventually, the overall discussions need to be improved.

2. There seems to be no phase transition from the heat capacity. Could the authors explain why the κ become less anisotropic at above RT compared to low-T.

3. The diffusivity vs temperature plot from laser flash is missing.

4. Is spin-orbit coupling included in the simulations with the presents of Bi? Will it affect the electronic structure, the phonon frequency or help stabilize the phonon dispersion beyond the current method of displacing atoms with the G-point eigenvectors?

5. Please improve the figure quality of Fig.3A inset. I couldn't distinguish the contribution from Cs-I and Cs-Cl. Yet, the areas under the curves seem to match the relative bonding strength.

6. Please improve the figure quality of Fig.3E.

7. Is the calculated bulk modulus valid in the presents of phonon instability?

8. In the Raman measurements, the authors identify 4 peaks below 80 cm^{-1} . A_{1g}(50), E_g(58), E_g(63), A_{1g}(80). A_{1g}(50) mode has large phonon softening, which is a characteristic of anharmonicity. However, the frequencies of the other three peaks are not shifted much from 4 – 300K, which contrary to the large Grüneisen parameter calculated from QHA in Fig.3C, reveals the limitation the QHA. The anharmonicity is reflected well from the increased Raman linewidth (enhanced scattering), which needs to be highlighted more.

9. There are a few papers discussing the anharmonic phonons related to another halide perovskite CsPbBr₃ that I'm aware of, which might help the authors to improve the discussion of the low kL in this compound.

<https://doi.org/10.1038/s41567-019-0520-x>

<https://doi.org/10.1038/s41563-021-00947-y>

<https://doi.org/10.1103/PhysRevLett.118.136001>

10. The ultralow kL is qualitatively ok with these discussions, however the abnormal temperature dependence of kL in Cs₃Bi₂I₆Cl₃ (40-150K) is not sufficiently explained in the current version.

Response to the Reviewers' comments

Reviewer #1

General Comment. *In this paper, Acharyya et al. report an ultralow (0.20 W/m.K) thermal conductivity and glass-like temperature dependence in a layered perovskite material $\text{Cs}_3\text{Bi}_2\text{I}_6\text{Cl}_3$. The authors grew single crystal samples, measured their thermal conductivities over a broad temperature range (2–400 K) using two methods, and performed theoretical calculations on chemical bonding and lattice dynamics. Moreover, the origin of soft phonon modes was tracked down to the rattling vibration of Cs^+ , by combining Raman and X-ray PDF refinement techniques. Overall, the materials performance presented herein is outstanding, the characterization results are comprehensive, and the discussions are thorough. I think this work will attract broad attention in research areas of thermoelectrics and perovskite materials, and recommend its publication in Nat. Comm. after considering the following suggestions and comments:*

Response. We would like to thank the reviewer for appreciating the work and recommending it for publication in Nature Communications. We have addressed all the concerns below.

Comment 1. *Sections and subheadings can be added to better organize the text contents.*

Response. We thank the reviewer for this comment. Now we have added sections and subheadings in the revised manuscript. Please see page numbers 1, 2, 5, 6, 8, 11, and 14.

Comment 2. *The special role of Cl⁻ in the soft phonon modes and ultralow thermal conduction can be investigated more. As mentioned in the text, 'charge clouds of Cl do not have any overlap with that of Bi' and 'the Bi-Cl bond distance is 2.92 Å whereas Bi-I bond distance is 2.90 Å in the global structure which is unusual'. Along with force constant calculations in Fig. 3A and potential curves in Fig. S7A, the Bi-Cl interaction can be better interpreted as ionic, and $\text{Cs}_3\text{Bi}_2\text{I}_6\text{Cl}_3$ can be understood as BiI_3 molecules distributed in an ionic matrix of Cs^+ and Cl^- , better than $[\text{Bi}_2\text{I}_6\text{Cl}_3]^{3-}$ layers (p. 6). The*

contrast between covalent and ionic bonds as well as between heavy and light atoms may well be a reason for the soft and largely isotropic phonon modes.

Response. We thank the reviewer for this very insightful comment. The ideal Bi-Cl ionic bond distance is 2.98 Å (ionic radii of Bi³⁺ and Cl⁻ are 1.17 and 1.81 Å respectively) whereas the covalent bond distance is 2.50 Å (covalent radii of Bi and Cl are 1.48 and 1.02 Å respectively). However, the Bi-Cl bond distance in Cs₃Bi₂l₆Cl₃ is 2.92 Å. In Cs₃Bi₂l₆Cl₃ crystal structure, Cl atoms are present in the bridging position of the [BiCl₄l₂]³⁻ octahedra, the Bi-Cl bond length is larger than normal covalent bond. This gives the distinct 2D structural characteristics to Cs₃Bi₂l₆Cl₃ in comparison to 0D nature of Cs₃Bi₂l₉. Thus, from the perspective of negligibly small charge cloud overlap between Bi and Cl and because of their weak bond stiffness, the Cs₃Bi₂l₆Cl₃ structure can also be alternatively described as Bil₃ molecules distributed in an ionic matrix of Cs⁺ and Cl⁻. We have added this revised description in page 8 and 9 (see revised manuscript, highlighted in yellow color). The presence of this bonding hierarchy with mixed ionic and covalent character and different masses of atom is indeed one of the reasons for the anharmonic lattice vibration and low κ_L in Cs₃Bi₂l₆Cl₃. Further, Bi-Cl interactions maintain anti-bonding character down to -2 eV below the Fermi level (E_f). Such filled anti-bonding states in the valence band just below the Fermi level soften the lattice (COHP, Fig 3A), leading to low thermal conductivity. We have also pointed out this aspect in the manuscript. Please see page 8-9 in the revised manuscript.

We have also emphasized on the consequent special role of Cl⁻ in the soft phonon modes and low thermal conductivity throughout the manuscript. The presence of Cl atom in the crystal structure is the one of the reasons for glass-like and ultralow thermal conductivity in Cs₃Bi₂l₆Cl₃. Due to its weak bonding environment, the atomic displacement parameter (ADP) of Cl is very high (Fig. S17, SI). Moreover, the site occupancy refinement from the synchrotron X-ray PDF data indicates Cl vacancy in the material [Occ (Cl): 0.957] which are known to hinder phonon transport through the defect scattering of phonons. In addition to PDF analysis, theoretical phonon dispersion also shows that the torsional mode distorts the soft [BiCl_{6/2}l₃]³⁻ octahedra due to the presence of a smaller size of Cl at the bridging position, making the crystal structure more flexible and elastically soft (Fig. S7, SI). We have discussed these aspects on page 9,10, and 13.

Comment 3. Several sentences should be double-checked for grammar:

a. p. 4: temperature dependence of κ_L in $\text{Cs}_3\text{Bi}_2\text{I}_6\text{Cl}_3$ is strongly deviates from...

b. p. 10: ... which could be seen from the presence of a Boson-like peak in C_p/T^3 vs. T plot (inset of Fig. 2C) and typical in amorphous materials and glasses.

c. p. 12: The peaks below ($r < 5 \text{ \AA}$) provide information about...

Response. We have corrected the unwarranted grammatical errors and improved the English in the revised manuscript.

Reviewer #2

General Comments. Crystalline solids with ultralow glass-like thermal conductivity have received great attention in recent years as they are important for example for thermoelectrics. This paper reports ultralow glassy thermal conductivity of a layered halide perovskite single crystal, $\text{Cs}_3\text{Bi}_2\text{I}_6\text{Cl}_3$. The authors have combined many techniques including Bridgman crystal growth, single crystal X-ray diffraction, Synchrotron X-ray Pair Distribution Function, DFT calculations, Raman spectra, and thermal conductivity characterizations, which is much appreciated. For a study reporting glassy thermal conductivity this paper is generally good, but the rationalization of glassy thermal conductivity is not well supported by the authors' claims and discussions. Also, the writing is a bit fragmented, and needs to be improved for a better connection between different sections. Below are some detailed comments for the authors to address before considering for publication.

Response. We would like to thank the reviewer for evaluating the manuscript and appreciating the work. We also understand the reviewer's legitimate concern about our claim and conclusion regarding the experimentally observed glass-like thermal conductivity of $\text{Cs}_3\text{Bi}_2\text{I}_6\text{Cl}_3$. We have tried to clarify this in our revised manuscript and addressed all the concerns with our point-by-point response below. We have revised the manuscript accordingly and improved our writing. Please see the highlighted sections in our manuscript. We hope that this revised manuscript would clarify all the concerns and acceptable for publication.

Comment 1. *The authors report the glassy ultralow thermal conductivity of $\text{Cs}_3\text{Bi}_2\text{I}_6\text{Cl}_3$ as well as many data proving large anharmonicity and local structural distortion, which is good. However, the reviewer did not see a clear connection between glassy ultralow thermal conductivity and large anharmonicity (or local structural distortion). One conclusion made by the authors is that localized anharmonic vibrations and local structural distortion result in glass-like temperature dependence of κ_L in $\text{Cs}_3\text{Bi}_2\text{I}_6\text{Cl}_3$. But what is the evidence to support this claim? The detailed origin of the glass-like temperature dependence probably requires further studies.*

Response. We thank the reviewer for appreciating our experimental and theoretical efforts to establish the presence of large anharmonicity and local structural distortion in $\text{Cs}_3\text{Bi}_2\text{I}_6\text{Cl}_3$. Now, concerning the connection between localized anharmonic vibration and local structural distortion to glass-like temperature dependence of κ_L , we would like to emphasize that our objective in this manuscript is to explore and establish the underlying microscopic phonon scattering mechanisms in this material. Subsequently, we relied on previous reports to conclude the effect of these phonon scattering mechanisms on κ_L and its temperature dependence. For example, we have shown clear evidence of anharmonic rattling like vibration (Fig. S9, S10 in SI) in $\text{Cs}_3\text{Bi}_2\text{I}_6\text{Cl}_3$. The connection between anharmonic rattling and glass-like κ_L has been well described in few earlier reports (Cryst. Res. Tech. 2017, 52, 1700114; Phys. Rev. B 2010, 81, 205207). In addition, thanks to your suggestions, we have now found that $\text{Cs}_3\text{Bi}_2\text{I}_6\text{Cl}_3$ has Cl vacancy and the connection between glass-like κ_L to such vacancy disordered is well established (Nat. Commun., 2021, 12, 6709). Thirdly, although we agree that the magnitude of local distortion is small, we established its presence in $\text{Cs}_3\text{Bi}_2\text{I}_6\text{Cl}_3$ beyond doubt using synchrotron X-ray PDF and Raman spectroscopy. In addition to PDF analysis, theoretical phonon dispersion also shows that the torsional mode distorts the soft $[\text{BiCl}_{6/2}]\text{I}_3^{3-}$ octahedra due to the presence of a smaller size of Cl at the bridging position, making the crystal structure more flexible and elastically soft (Fig. S7, SI). Again, few earlier papers have studied how such local distortion leads to glass-like κ_L (J. Appl. Phys. 2016, 119, 185102; Angew. Chem. Int. Ed. 2022, 61, e202200071; J. Am. Chem. Soc. 2020, 142, 15187). To reiterate, we have not explored the role of a specific phonon scattering mechanism leading to glass-like κ_L . Rather, we experimentally observed glass-like κ_L in $\text{Cs}_3\text{Bi}_2\text{I}_6\text{Cl}_3$ and using experimental and theoretical investigation we established the several dominant

phonon scattering mechanisms in Cs₃Bi₂I₆Cl₃. We established that soft lattice, large anharmonicity and low frequency optical phonon modes due to rattling, Cl vacancy and local structural distortion are the dominant phonon scattering mechanism in this material and therefore their combined presence should be responsible for the observed glass-like κ_L in Cs₃Bi₂I₆Cl₃. Please see page 13 in revised manuscript.

Comment 2. *Defects and disorders are common in halide perovskites. The structural analyses on single crystal X-ray diffraction, synchrotron powder X-ray diffraction, and Pair Distribution Function did not include the refinements of site occupancies. Detailed analysis of site occupancies is required given point defects are common in halide perovskites.*

Response. We thank the reviewer for this comment. We have now carried out the site occupancies refinement on single crystal X-ray diffraction data and the synchrotron X-ray Pair Distribution Function (PDF) data. We find that the refinement of site occupancies from room temperature single crystal data does not show any significant amount of vacancies present in the systems (Table S2, SI). We would like to point out that because of the presence of heavy atoms in the crystal, absorption at the measured wavelength ($\lambda = 0.71073 \text{ \AA}$) in the laboratory single crystal diffraction is quite high. Thus, in case of small vacancy concentration, it is quite challenging to convincingly demonstrate whether any vacancy is present in the system or not from laboratory X-ray diffraction.

However, the absorption is minimal at the wavelength (0.20742 \AA) using which we collected the synchrotron X-ray Pair Distribution Function (PDF) data. The refinement of synchrotron X-ray PDF data at 300 K indeed indicates Cl vacancy in the material convincingly. Please see the table below (Table R1). We have revised the manuscript also accordingly. Please see page 13 in revised manuscript.

Table R1. *Refinement of site occupancy using synchrotron X-ray PDF data at 300 K.*

Atom	Occupancy
Cs1	1.000
Cs2	1.003

Bi	1.001
I	0.992
Cl	0.957

Comment 3. *The phonon lifetime extracted from Raman spectra shows a decreasing trend with rising temperature, which reflects strong phonon scattering processes increasing with temperature. One may expect a decreasing Kappa with increasing temperature as the phonon lifetime decreases. This result and argument from the authors seem to be conflicting with glassy thermal conductivity data, which show a weak increasing trend with temperature. Can the authors comment on this?*

Response. We understand the reviewer's concern on phonon lifetime from Raman spectra. The increasing linewidth of the Raman peak with increasing temperature indeed indicate decrease in phonon lifetime. However, similar diminishing trends in the phonon lifetime have been found in other compounds such as AgSbTe₂ (Nat. Nanotech., 2013, 8, 445), AgSbSe₂ (Angew. Chem. Int. Ed. 2022, 61, e202200071) and clathrate Ba₈Ga₁₆Sn₃₀ (Phys. Rev. B, 2010, 81, 205207) which show glass-like thermal conductivity (i.e., κ_L is nearly temperature independent or slowly increases with increasing temperature). Raman spectroscopy probe only the optical phonon modes at Γ point in the Brillouin zone. On the other hand, thermal transport involves phonon modes (mainly acoustic phonons) over the entire Brillouin zone and as seen from our DFT calculation (Fig. 3D, E), multiple low energy optical phonon modes are present all over the Brillouin zone which can couple and scatter acoustic modes. Many of these phonon modes are highly anharmonic and it is possible that these modes may show temperature dependent hardening (e.g. Nat. Mater., 2008, 7, 811; Rev. Mod. Phys., 2014, 86, 669). This could be one explanation for the slowly increased lattice thermal conductivity with increasing temperature. The other probable reason could be the presence of *diffusons* in Cs₃Bi₂l₆Cl₃ as seen from the analysis of participation ratio of phonon modes (Figure S11, SI; e.g. J. Appl. Phys. 2016, 120, 025101; Adv. Mater. 2019, 31, 1808222). Unlike phonons, *diffusons* are not bounded by lattice periodicity and they propagate thermal energy via random walk (Energy Environ. Sci., 2018, 11, 609; NPJ Comput. Mater., 2017, 3, 49). Hence, the thermal energy is mainly transported via diffusive thermal transport by *diffusons*. The κ_L due to *diffuson* modes increases with temperature (NPJ Comput. Mater., 2017, 3, 49). Please see the

discussion in page 10 in revised manuscript. In conclusion, while we have presented a number of evidence behind the appearance of a glassy κ_L in $\text{Cs}_3\text{Bi}_2\text{I}_6\text{Cl}_3$, we agree that this requires further investigation (i.e., follow up works) which would highlight the similarities and differences between the observed glass-like thermal conductivity in $\text{Cs}_3\text{Bi}_2\text{I}_6\text{Cl}_3$ and those of typical glass and amorphous materials.

Comment 4. According to the point group of $D_{3d}(-3m)$ of $P-3m1$, Raman active modes include $4A_{1g} + 5E_g$. From the spectra in Fig. 4A, only $3A_{1g}$ and $3E_g$ modes are indexed. How about the other $1A_{1g}$ and $2E_g$ modes? Can the authors explain this?

Response. We thank the reviewer for pointing this out. Indeed, the group theoretical analysis give us $4A_{1g} + 5E_g$ Raman active modes in $\text{Cs}_3\text{Bi}_2\text{I}_6\text{Cl}_3$. Out of these 9 Raman active modes, we observed $3A_{1g}$ and $3E_g$ modes in Fig. 4A, which was measured using 785 nm laser excitation. We also observed one additional E_g mode at 38 cm^{-1} using 633 nm laser excitation with low frequency filters (Fig. S14, SI). The $1A_{1g}$ ($\sim 20 \text{ cm}^{-1}$) and $1E_g$ at ($\sim 22 \text{ cm}^{-1}$) are not visible because of their occurrence in very low frequency in which we do not have sensitivity of our Raman measurement. Rayleigh tail is large below 30 cm^{-1} and we could not observe the phonon modes below this frequency range. Please see the list of theoretically expected Raman active modes and comparison to our experimental observation in $\text{Cs}_3\text{Bi}_2\text{I}_6\text{Cl}_3$ in Table R2 below. We have also revised the manuscript accordingly. Please see page 11-12, in revised manuscript.

Table R2. List of theoretically expected and experimentally observed Raman modes in $\text{Cs}_3\text{Bi}_2\text{I}_6\text{Cl}_3$.

S.N	Frequency (cm^{-1}) (Theoretical)	Mulliken symbol	Frequency (cm^{-1}) (Experiment)		
			785 nm laser		633 nm laser at 300 K
			At 4 K	At 300 K	
1	19.6	A_{1g}	Not observed	Not observed	Not observed
2	21.6	E_g	Not observed	Not observed	Not observed

3	37.9	E_g	Not observed	Not observed	38 (with low frequency filter)
4	52.5	A_{1g}	51.3	48	Not observed
5	54.1	E_g	58.9	58	56.1
6	65.5	E_g	65.3	63	65
7	75.2	A_{1g}	77.7	75	75.7
8	132.8	E_g	136.4	132	132.6
9	151.1	A_{1g}	150.8	147	151.2

Comment 5. *From the PDF analysis, it seems that the local distortion is quite small as the change in bond distance is rather small (at the second decimal). The reviewer is a bit suspicious of how such a small distortion in the bond distance would result in a glass-like temperature dependence of Kappa. Can the authors elaborate in detail on how a local structural distortion would result in enhanced phonon scattering strength and a glass-like temperature dependence of Kappa? A poor fit of the synchrotron X-ray PDF data usually indicates structure disorder. The refinement model based on the reported cif file did not consider defects such as interstitials and vacancies, which might be the origin of the glass-like Kappa. Especially, static disorder is often considered as an important reason for the glass-like Kappa. The authors could refer to a recent paper (Nat Commun 12, 6709 (2021)) for details on studying subtle structure disorders in simple crystalline structure using advanced crystallographic methods.*

Response. We understand the reviewer's concern and we thank reviewer for this very good suggestion. Now, we have carried out refinement for site occupancies using synchrotron X-ray PDF data, which indeed indicates the presence of a Cl vacancy in $Cs_3Bi_2I_6Cl_3$ (please see response for comment #2 and Table R1). We also thank the reviewer for suggesting to look into the recent work in Nat. Commun., 2021, 12, 6709,

which helped us to understand better and we have cited it in our revised manuscript (Reference number 60). The presence of this Cl vacancy indeed enhances phonon scattering and it should contribute to the observed glass-like thermal conductivity in $\text{Cs}_3\text{Bi}_2\text{I}_6\text{Cl}_3$ as an additional factor. Please see page 13 in revised manuscript.

We agree that the magnitude of local distortion is rather small in $\text{Cs}_3\text{Bi}_2\text{I}_6\text{Cl}_3$. However, the presence of a such a local distortion in the Bi-halide octahedra lowers the local structural symmetry which can lead to glass-like κ_L and it was previously observed in few compounds (J. Appl. Phys. 2016, 119, 185102; Angew. Chem. Int. Ed. 2022, 61, e202200071; J. Am. Chem. Soc. 2020, 142, 15187). In addition to PDF analysis, theoretical phonon dispersion also shows that the torsional mode distorts the soft $[\text{BiCl}_{6/2}\text{I}_3]^{3-}$ octahedra due to the presence of a smaller size of Cl at the bridging position, making the crystal structure more flexible and elastically soft (Fig. S7, SI). We emphasize that it is the combined presence of soft crystalline lattice resulting in soft acoustic modes, highly anharmonic lattice vibration with the presence of rattler-like phonon modes, and the local structural distortion in the $[\text{BiCl}_3\text{I}_3]^{3-}$ octahedra which results in the glass-like thermal conductivity in $\text{Cs}_3\text{Bi}_2\text{I}_6\text{Cl}_3$. The presence of Cl vacancy, as we found from synchrotron X-ray PDF refinement, should also add into these effects enhancing the phonon scattering. The intriguing aspect of our study is that the underlying reason for the presence of these multiple strong phonons scattering channels is the hierarchical and unique chemical bonding in this halide perovskite. We have revised the manuscript also accordingly. Please see page 13 in revised manuscript.

Finally, we would also like to mention that the glass-like thermal conductivity in $\text{Cs}_3\text{Bi}_2\text{I}_6\text{Cl}_3$ should be fundamentally different from those observed in conventional disordered and amorphous materials. Because of the presence of multiple strong phonon scattering mechanisms involving local structural distortion, Cl vacancy and lattice anharmonicity due to rattling, probably neither the harmonic theory of disordered materials (like Cahill Watson and Pohl or Allen-Feldman theory) or Peierls-Boltzmann transport theory (Energy Environ. Sci., 2018, 11, 609) would be sufficient to describe the experimentally observed glass-like thermal conductivity in $\text{Cs}_3\text{Bi}_2\text{I}_6\text{Cl}_3$. This is an ideal opportunity where the newly developed unified theory of thermal transport should bring in more insight (Nat. Phys. 2019, 15, 809). However, this is right

now out of scope for this manuscript, and we strive to address this in the future follow up works. We hope the reviewer would understand this.

Comment 6. *Can the authors comment on the origin of the unstable phonon modes in phonon calculations? Are they caused by the structure disorder or strong anharmonicity? Can these unstable phonons be removed by applying hydrostatic pressure or a phonon renormalization scheme?*

Response. The presence of unstable phonon modes in the harmonic (i.e., $T = 0$ K) phonon dispersion could originate from a variety of reasons. For example, density functional theory (DFT) calculations are performed at $T = 0$ K utilizing a crystal structure that is usually refined at a finite temperature ($T \neq 0$ K) in experiment. Hence, the $T = 0$ K crystal structure of the compound can be slightly different from its $T \neq 0$ K structure. Thus, the use of $T \neq 0$ K crystal structure in $T = 0$ K DFT calculations could lead to unstable phonon modes in the calculated phonon dispersion. However, our low temperature experiment down to 2 K (heat capacity and thermal conductivity) does not reveal the presence of a structural phase transition in $\text{Cs}_3\text{Bi}_2\text{I}_6\text{Cl}_3$. Hence, this cause is ruled out in this case.

Another reason could be the absence of anharmonic interactions between the phonons in the $T = 0$ K harmonic phonon calculations. Since it is well known that halides perovskites and their derivative compounds are strongly anharmonic (J. Phys. Chem. Lett. 2017, 8, 2659; Nat. Mat., 2021, 20, 977; Adv. Mater., 2022, 34, 2107932), the inclusion of anharmonic interactions through phonon renormalization at finite temperatures could stabilize the unstable harmonic phonon modes. Since calculations of the anharmonic phonon renormalization is computationally very expensive, this is beyond the scope of the present work. We strongly believe this is the possible origin of the negative phonon modes in our calculation as the $\text{Cs}_3\text{Bi}_2\text{I}_6\text{Cl}_3$ is intrinsically anharmonic system.

Sometimes, unstable phonon could be stabilized by applying hydrostatic pressure or isotropic strain. We applied 6% compressive isotropic volume strain on the unit cell of $\text{Cs}_3\text{Bi}_2\text{I}_6\text{Cl}_3$ and relaxed its internal coordinates of the compound using DFT calculations. We calculated the harmonic dispersion of this compressed and relaxed unit cell which shows partial stabilization of the unstable phonon modes (Fig.

R1). We have added this discussion in the computational methods section. Please see page number S8, SI and a new Fig. S20.

Fig. R1. Phonon dispersion of $\text{Cs}_3\text{Bi}_2\text{I}_6\text{Cl}_3$ for the **(A)** equilibrium crystal structure (i.e., 0 % strain) and **(B)** for the strained crystal structure, where volume is compressed isotropically by 6%.

Comment 7. *The reviewer has a question on the Bridgman grown crystal. As shown in Fig. 1D, in principle, one may expect the $(hk0)$ reflections for the // Bridgman direction if it is a perfect single domain. It is usually difficult to ensure a single domain for the whole large crystal. Can the authors comment on this? Could it be a misalignment during cutting or the crystal being multi-domain?*

Response. We understand the reviewer's concern. We agree that it is difficult to ensure a single domain for the such a large crystal. We have observed that all the $(h k 0)$ planes [see, (2-10), (3-10) and (4-20)] along with the other $(h 0 l)$ peaks for parallel to Bridgman direction within the measured 2-theta. This is possibly because of small misalignment which leads to the appearance of other, particularly $(h 0 l)$ peaks, as $(h k 0)$ and $(h 0 l)$ planes have small angle between them (please see the figure below). We must say that cutting the crystal is extremely difficult as it falls apart most of the time due to cleavage planes but we got success few times. The good quality of the single crystal is ascertained by observation of predominantly $(00 l)$ for perpendicular to the Bridgman direction and single crystal X-ray data.

Fig. R2. Illustration of $(0\ 0\ l)$, $(h\ k\ 0)$ and $(h\ 0\ l)$ planes in $\text{Cs}_3\text{Bi}_2\text{I}_6\text{Cl}_3$. Cs, Bi, I, and Cl atoms are denoted by the blue, red, yellow, and green spheres, respectively.

Reviewer #3

General Comment. *This work studies the origin of the glassy-like thermal conductivity in all-inorganic metal halide perovskite $\text{Cs}_3\text{Bi}_2\text{I}_6\text{Cl}_3$ single crystals with a combined experimental and theoretical approach. The single crystal has high quality with size being sufficiently large to give promising k_L . The k_L is surprisingly low $0.2\ \text{W/mK}$ at $300\ \text{K}$, and the plateau behavior in k_L is indeed interesting and uncommon compared to many other inorganic halide perovskites. In summary, the reviewer thinks the work is interesting, but there are a few technical questions that need to be addressed:*

Response. We are thankful to the reviewer for appreciating our work. We are also thankful for reviewer's insightful comments and suggestions.

Comment 1. *The experimental observation in κ_L is interesting and sound. However, the simulation part based on quasi harmonic approximation may not be sufficient to fully account for the unusual κ_L . Even if anharmonic calculations (e.g., AIMD) or phonon scattering rate calculations will not be included eventually, the overall discussions need to be improved.*

Response. We thank the reviewer for this comment and suggestion with which we completely agree. We have added additional details in the revised manuscript on possible failure of the (quasi) harmonic approximation to describe the glass-like κ_L of $\text{Cs}_3\text{Bi}_2\text{I}_6\text{Cl}_3$, which is strongly anharmonic. Quasi harmonic approximation (QHA) is widely used to analyse the lattice dynamics of materials that albeit provides many useful physical information about those compounds. For example, the $T=0\text{K}$ phonon calculations of $\text{Cs}_3\text{Bi}_2\text{I}_6\text{Cl}_3$ already exhibits intriguing features in its phonon dispersion such as local distortion of the mixed halide octahedra around Bi, weakly dispersive soft phonon branches, and rattling vibrations of Cs atoms. However, we acknowledge that QHA is not sufficient to provide a more quantitative description of the phonons and their changes caused by anharmonic interactions at finite temperatures. It was shown that for strongly anharmonic compounds such as Tl_3VSe_4 (Phys. Rev. Lett., 2020, 124, 065901) and TlInTe_2 (NPJ Comput. Mater., 2021, 7, 82), the phonon scattering rates calculated utilizing the harmonic phonons fails completely to describe the measured κ_L in those compounds, necessitating the inclusion of the anharmonic effects on the phonons through the calculations of phonon self-energies. Recent experiments and theoretical studies (Phys. Rev. Lett. 2017, 118, 136001; Nat. Phys., 2019, 15, 809; Nat. Mater., 2021, 20, 977) also reveal that halide perovskites are extremely anharmonic in nature, manifesting damped vibrations of the octahedral motions, soft phonons, and extremely short phonon lifetimes. Thus, to capture such strong anharmonic effects at finite temperatures, ab initio molecular dynamics (AIMD) simulations would be necessary to renormalize the phonon frequencies and estimate their lifetimes accurately. AIMD simulations have been performed recently on cubic halide perovskite (Nat. Mater., 2021, 20, 977) and double perovskite (Phys. Rev. Lett. 2020, 125, 045701), possessing very high symmetry and small number of atoms in their unit cells (≤ 10). Since AIMD simulations are computationally very expensive, studying the temperature dependent lattice dynamics of $\text{Cs}_3\text{Bi}_2\text{I}_6\text{Cl}_3$ (having hexagonal symmetry and 14 atoms in the unit cell) including full anharmonicity would be an interesting yet challenging theoretical task. We have added new discussion to the revised manuscript in page 14.

Comment 2. *There seems to be no phase transition from the heat capacity. Could the authors explain why the κ_L become less anisotropic at above RT compared to low- T .*

Response. In general, anisotropy in κ_L of two-dimensional (2D) materials decreases with increasing temperature (SnSe: Science, 2015, 351, 141; Cs₂PbI₂Cl₂: J. Am. Chem. Soc., 2020, 142, 15595; PdSe₂: Mater. Today Phys. 2022, 22, 100599). Thus, our observation of decreasing anisotropy in κ_L with increasing temperature (above 300 K) is in accordance with the reports of other 2D materials. One of the main reasons for this decrease in anisotropy in κ_L with increasing temperature is that the constituent atoms vibrate with greater amplitude as temperature increases. This enhances overall bonding isotropy of the system and consequently, κ_L becomes less anisotropic at higher temperature. Moreover, as seen from the chemical bonding analysis (page 7 and Fig. S5, SI), the interlayer separation in Cs₃Bi₂I₆Cl₃ is much lower compared to the ideal van der Waals (vdW) bond. As a result, the thermal anisotropy is already very weak in Cs₃Bi₂I₆Cl₃.

Please also note the thermal conductivity has been measured by two independent methods in the different temperature range: (a) At above the room temperature, it is measured by laser flash diffusivity technique where the accuracy is higher at higher temperatures and (b) In the 2-300 K, it is measured by PPMS, where in lower temperatures, the accuracy is excellent, but, near room temperature, the measured thermal conductivity has large error bar due to radiation leakage. It is worth to be mentioned that at room temperature, data from two techniques converges well within the error bar.

Comment 3. *The diffusivity vs temperature plot from laser flash is missing.*

Response: We thank the reviewer for this comment. Now we have added the diffusivity vs. temperature plot (300-400 K) plot in the revised SI (Please see Fig. S3, SI).

Comment 4. *Is spin-orbit coupling included in the simulations with the presents of Bi? Will it affect the electronic structure, the phonon frequency or help stabilize the phonon dispersion beyond the current method of displacing atoms with the G-point eigenvectors?*

Response. We have calculated the electronic structures of $\text{Cs}_3\text{Bi}_2\text{I}_6\text{Cl}_3$ without and with the inclusion of the spin-orbit coupling (SOC), which reveals that SOC reduces the band gap from 1.94 eV (Fig. R3A) to 1.31 eV (Fig. R3B) and splits the degeneracy of the bands along low-symmetry directions in the Brillouin zone. On the other hand, SOC has negligible effect on the phonon dispersion, and it does not help in stabilizing the unstable phonon modes (Fig. R3C, D). We have added this section to the revised supporting in page S8 and added a new Fig. S21, SI.

Fig. R3. Electronic structures (A, B) and phonon dispersions (C, D) of $\text{Cs}_3\text{Bi}_2\text{I}_6\text{Cl}_3$ calculated without and with the inclusion of spin-orbit coupling (SOC).

Comment 5. Please improve the figure quality of Fig.3A inset. I couldn't distinguish the contribution from Cs-I and Cs-Cl. Yet, the areas under the curves seem to match the relative bonding strength.

Response. We have changed the colours for Cs-I and Cs-Cl interactions and also provided a zoomed in version (Fig. R4). Both Cs-I and Cs-Cl have small interactions strength, which is evident from the crystal orbital Hamilton population (COHP) analysis and as well as force constants plots (Fig. 3A). Please see new figure in Fig. S6, SI in revised manuscript.

Fig. R4. Crystal orbital Hamilton population (COHP) analysis (**A**) in the range of -2.0 to 2 eV/cell and (**B**) -0.2 to 0.2 eV/cell.

Comment 6. Please improve the figure quality of Fig.3E.

Response. We thank the reviewer for this comment. We have improved the figure quality in the revised manuscript. Please see the revised manuscript, page no. 26.

Comment 7. Is the calculated bulk modulus valid in the presents of phonon instability?

Response. Yes, the calculated bulk modulus is valid in the presence of phonon instability. The bulk modulus is obtained using the Voigt-Reuss-Hill (VRH) formula (Proc. Phys. Soc. A 1952, 65, 349) that utilizes the second order elastic constants. The second order elastic constants are obtained by applying homogeneous and infinitesimal strain to the crystal at equilibrium and subsequently by polynomial fitting of the energy of the deformed crystal as a function of strain.

We have also estimated the bulk and shear modulus from experimental sound velocity which agrees well with the theoretically calculated elastic moduli as shown in

the Table R3. We have included this table in the supplementary information. Please see Table S4 on the page S33 in revised manuscript.

Table R3. Bulk and shear modulus of Cs₃Bi₂l₆Cl₃

Elastic moduli	Theoretical calculation (GPa)	Experimental estimation (GPa)	
		Cs ₃ Bi ₂ l ₆ Cl ₃ (\perp Bridgman)	Cs ₃ Bi ₂ l ₆ Cl ₃ (\parallel Bridgman)
Bulk	14.4	15.2	14.7
Shear	6.7	4.2	4.9

Comment 8. *In the Raman measurements, the authors identify 4 peaks below 80 cm⁻¹. A_{1g} (50), E_g (58), E_g (63), A_{1g} (80). A_{1g} (50) mode has large phonon softening, which is a characteristic of anharmonicity. However, the frequencies of the other three peaks are not shifted much from 4–300 K, which contrary to the large Grüneisen parameter calculated from QHA in Fig.3C, reveals the limitation the QHA. The anharmonicity is reflected well from the increased Raman linewidth (enhanced scattering), which needs to be highlighted more.*

Response. We understand the reviewer’s concern and also appreciate the suggestion about the Raman linewidth. Indeed, the A_{1g} (~ 48 cm⁻¹) mode has the highest phonon softening among these four peaks. However, other three peaks also show temperature dependent softening (please see Fig. R5). We find that the A_{1g} mode at 48 cm⁻¹ and 75 cm⁻¹ show higher temperature dependent phonon softening compared to that of the E_g modes at 58 cm⁻¹ and 63 cm⁻¹. Interestingly, all these four phonon modes involve vibration of only Cl atoms (Fig. S13, SI). Although we agree that QHA has its own limitation, the comparison of our calculated mode Grüneisen parameter (Fig. 3C) with temperature dependent Raman data does not show any immediate discrepancies. Firstly, Fig. 3C exhibits mode Grüneisen parameter, which was calculated by excluding the phonon modes at Γ point and in the small vicinity of Γ point as for very small phonon frequencies, the mode Grüneisen parameter tends to diverge. Whereas,

the Raman peaks essentially represent only the phonon modes at the Γ point. Secondly, theoretical calculation exhibits highest mode Grüneisen parameter between $0 - 30 \text{ cm}^{-1}$, in which range we have many low frequency optical modes throughout the entire Brillouin zone (which contributed to the calculated mode Grüneisen parameter shown in Fig. 3C), whereas our Raman measurement is sensitive (in present case) only over 30 cm^{-1} (in general, Rayleigh tail is large below 30 cm^{-1}). Thus, we could not probe these phonon modes below 30 cm^{-1} using Raman spectroscopy, which could show much larger temperature dependent phonon softening. While Fig. 3c show mode Grüneisen parameter is moderately higher in the $30\text{-}50 \text{ cm}^{-1}$ compared to that of the $>50 \text{ cm}^{-1}$, which probably explains more temperature dependent softening of A_{1g} ($\sim 48 \text{ cm}^{-1}$) mode. We have further revised and highlighted the discussion on the increased phonon linewidth with increasing temperature and its correlation with scattering time in the manuscript. Please see the highlighted text in page 12, 14-15 in the revised manuscript. Please also see Fig. 4B and S14 and S15, SI.

Fig. R5. The temperature dependent peak shift for all the four Raman modes in the frequency range below 80 cm^{-1} exhibiting softening of the modes.

Comment 9. *There are a few papers discussing the anharmonic phonons related to another halide perovskite CsPbBr_3 that I'm aware of, which might help the authors to improve the discussion of the low k_L in this compound.*

<https://doi.org/10.1038/s41567-019-0520-x>

<https://doi.org/10.1038/s41563-021-00947-y>

<https://doi.org/10.1103/PhysRevLett.118.136001>

Response. We thank the reviewer to suggest these papers. We have duly cited these papers and added discussion in the revised manuscript. Please see page 14 and reference no. 42, 43 and 48 in revised manuscript.

Comment 10. *The ultralow k_L is qualitatively ok with these discussions, however the abnormal temperature dependence of k_L in $\text{Cs}_3\text{Bi}_2\text{I}_6\text{Cl}_3$ (40-150 K) is not sufficiently explained in the current version.*

Response. We understand the reviewer's concern. Crystalline materials generally exhibit an $\sim T^3$ dependent lattice thermal conductivity (κ_L) at low temperature, followed by a peak after which κ_L decreases following a T^{-1} dependence due to Umklapp scattering (Please see the page 7-8 in revised manuscript). Therefore, it is obvious that temperature dependence of κ_L in $\text{Cs}_3\text{Bi}_2\text{I}_6\text{Cl}_3$ in this temperature regime (40 -150 K) is significantly different from a typical crystalline material. Temperature dependent κ_L in $\text{Cs}_3\text{Bi}_2\text{I}_6\text{Cl}_3$ in 40 – 150 K is more similar to glassy and amorphous materials. This regime in glassy materials appears above the tunnelling dominated thermal transport ($\kappa_L \sim T^2$) and have nearly temperature independent κ_L because of Rayleigh-type scattering originating from the atomic scale positional disorder of the atoms (Cryst. Res.Tech. 2017, 52, 1700114). The product of phonon mean free path and heat capacity becomes constant in this regime leading to temperature independent κ_L . Although, the atomic scale positional disorder is not immediately evident from our experiment, we find evidence of local structural distortion and Cl vacancy in $\text{Cs}_3\text{Bi}_2\text{I}_6\text{Cl}_3$ by analysing the synchrotron X-ray PDF. Few earlier papers have studied how such local distortion and vacancy like defects leads to glass-like κ_L (J. Appl. Phys. 2016, 119, 185102; Angew. Chem. Int. Ed. 2022, 61, e202200071; J. Am. Chem. Soc. 2020, 142, 15187; Nat. Commun., 2021, 12, 6709). Further, the atomic displacement parameters (ADPs) are found to be high, mainly for Cs and to an extent for Cl as well. The high ADP for Cs resembles its rattling character. The connection between anharmonic rattling and glass-like κ_L has well been described in few earlier reports

(Cryst. Res.Tech. 2017, 52, 1700114; Phys. Rev. B 2010, 81, 205207). To reiterate, we have not explored the role of a specific phonon scattering mechanism leading to glass-like κ_L . Rather, we experimentally observed glass-like κ_L in $\text{Cs}_3\text{Bi}_2\text{I}_6\text{Cl}_3$ and using experimental and theoretical investigation we established the several dominant phonon scattering mechanisms in $\text{Cs}_3\text{Bi}_2\text{I}_6\text{Cl}_3$. We established that soft lattice, large anharmonicity and low frequency optical phonon modes due to rattling, Cl vacancy and local structural distortion are the dominant phonon scattering mechanisms in this material and therefore their combined presence should be responsible for the observed glass-like κ_L in $\text{Cs}_3\text{Bi}_2\text{I}_6\text{Cl}_3$. Please see page 13 in revised manuscript.

We agree that understanding this temperature regime requires further investigation, possibly using inelastic X-ray and neutron scattering. We indeed like to do these measurements for better understanding in future. We hope the reviewer would understand this limitation at this moment.

REVIEWERS' COMMENTS

Reviewer #1 (Remarks to the Author):

This manuscript is a revised version of NCOMMS-22-15883, which I reviewed before. The authors have performed new experiments and theoretical calculations, as well as additional analyses and discussions of the results, to address the concerns and comments from all reviewers, especially about the anharmonic vibrational modes. I'm satisfied with the authors' efforts to enhance the quality of this manuscript, and think that the current version meets the standards of novelty, significance, and completeness for publication in Nat. Commun.

Reviewer #2 (Remarks to the Author): no response

Reviewer #3 (Remarks to the Author):

The authors have addressed my comments and the discussions are improved.

Response to the Reviewers' comments

Reviewer #1

General Comment. *This manuscript is a revised version of NCOMMS-22-15883, which I reviewed before. The authors have performed new experiments and theoretical calculations, as well as additional analyses and discussions of the results, to address the concerns and comments from all reviewers, especially about the anharmonic vibrational modes. I'm satisfied with the authors' efforts to enhance the quality of this manuscript and think that the current version meets the standards of novelty, significance, and completeness for publication in Nat. Commun.*

Response. We would like to thank the reviewer for appreciating the work and recommending it for publication in Nature Communications.

Reviewer #2

General Comments. *No response*

Response. Not applicable.

Reviewer #3

General Comment. *The authors have addressed my comments and the discussions are improved.*

Response. We are thankful to the reviewer for appreciating our work.